# Enantiomer-selective magnetization of conglomerates for quantitative chiral separation

Xichong Ye [1,4], Jiaxi Cui[2,3,4], Bowen Li[1], Na Li[1], Rong Wang[1], Zijia Yan[1], Junyan Tan[1], Jie Zhang[1] & Xinhua Wan [1]

Selective crystallization represents one of the most economical and convenient methods to provide large-scale optically pure chiral compounds. Although significant development has been achieved since Pasteur's separation of sodium ammonium tartrate in 1848, this method is still fundamentally low efficient (low transformation ratio or high labor). Herein, we describe an enantiomer-selective-magnetization strategy for quantitatively separating the crystals of conglomerates by using a kind of magnetic nano-splitters. These nano-splitters would be selectively wrapped into the $S$-crystals, leading to the formation of the crystals with different physical properties from that of $R$-crystals. As a result of efficient separation under magnetic field, high purity chiral compounds (99.2 ee% for $R$-crystals, 95.0 ee% for $S$-crystals) can be obtained in a simple one-step crystallization process with a high separation yield (95.1%). Moreover, the nano-splitters show expandability and excellent recyclability. We foresee their great potential in developing chiral separation methods used on different scales.

[1] Beijing National Laboratory for Molecular Sciences, Key Laboratory of Polymer Chemistry and Physics of Ministry of Education, College of Chemistry and Molecular Engineering, Peking University, Beijing 100871, China. [2] INM - Leibniz Institute for New Materials, Campus D2 2, 66123 Saarbrucken, Germany. [3] Institute of Fundamental and Frontier Sciences, University of Electronic Science and Technology of China, Chengdu 611731, China. [4] These authors contributed equally: Xichong Ye, Jiaxi Cui. Correspondence and requests for materials should be addressed to X.W. (email: xhwan@pku.edu.cn)

People have never stopped seeking for more efficient methods to generate optically pure compounds to meet the huge demand in modern pharmaceutics, since Louis Pasteur separate the crystals of enantiomers by using a magnifier and tweezers[1–3]. These methods currently available are mainly based on two strategies, i.e., asymmetric synthesis[4] and chiral resolution. Compared to the former, chiral resolution still plays the major role in producing chiral drugs due to its high reliability and wide applicability in industry. Racemates can be separated by selective crystallization, enzyme[5], and chromatography etc[6,7]. Undoubtedly, selective crystallization is still the most economical and convenient method to provide large-scale chiral compounds[8]. Through 170-year development of this method, several sub-strategies have been developed, i.e., spontaneous crystallization[9], diastereomer crystallization[10,11], preferential crystallization[12] (including heteronucleation assisted by nanoparticles[13,14]) and tailor-made additives for selective inhibition[15–17], etc. Some of these strategies are quite sophisticated in providing mass optically pure enantiomers through multiple-cycle crystallization assisted by complex industrial setups. Methods like crystallization-induced asymmetric transformation[18], and Viedma ripening[19,20] involving a racemization process can obtain optically pure crystals with 100% yield. Despite of these progresses, the fundamental challenges in selective crystallization, i.e., directly, easily separating the pure enantiomer crystals formed in a simple one-step crystallization process with quantitative chemical yield, without requiring chemical transformation, have never been overcome. Note that the enantiomeric crystals of conglomerates are intrinsically enantiomeric pure, even precipitate together. The difficulty originates from distinguishing and separating these tiny crystals which normally have the same size and mirror geometry.

Herein, we present a strategy for quantitatively collecting optically pure compounds from conglomerates by selectively embedding magnetic nanoparticles (MNPs) into one enantiomer's crystals (Fig. 1a). The magnet-responsive strategy has been wildly applied in many fields[21–23], including magnetic separation used in beneficiation[24]. Recently, electron spin orientation of magnetic material is suggested to be enantiospecific preference and can be applied for chiral resolution[25]. However, this method is inefficient (low yield and ee%) because an adsorption strategy was used to enrich the molecules with same electron spin orientation. Our method is making one enantiomeric crystal magnetic-responsive by embedding MNPs into it and leaving the other non-magnetic. Thus, large-scale self-sorting can be realized under a magnetic field. This simple strategy is fundamentally different from traditional selective crystallization that leads to enantiomorphs with same physical properties (Fig. 1a). For selective embedding the MNPs into one of the enantiomorphs, specific interaction should be built up between the nanoparticles and the target enantiomer. To this end, we added both seeds to accelerate the crystallization of one enantiomer and magnetic inhibitors to delay and mark the nucleation of the other enantiomer in the oversaturated racemic solution. This method was expected to differ the crystallization kinetics of the enantiomers, to generate the R-crystals and S-crystals separately, avoiding the adhesion between enantiomeric crystals. The inhibiting effect would further promote a local Ostwald ripening process in which the inhibitors were wrapped by the same enantiomer to grow into magnetic polycrystals in the later stage of the crystallization[26]. As a result, self-sorting of enantiomeric crystals can be realized by using this kind of nanoscopic inhibitors, which we would like to call them nano-splitters. This strategy was expected to allow for efficient separation (high yield) of optical pure products (high ee% values) in a low cost, recyclable way for different scales.

## Results

**Synthesis of magnetic nano-splitters**. Based on this concept, we designed and synthesized a class of magnetic nano-splitters consisting MNPs as the core and the polymeric inhibitors with high stereoselectivity to the target crystals as the shell. The nano-splitters were made by co-assembling amphiphilic poly($N^6$-methacryloyl-S-lysine)-block-polystyrene ((S)-PMAL$_m$-b-PSt$_n$) diblock copolymers and hydrophobic $Fe_3O_4$@oelic acid MNPs (Fig. 1b). The copolymers (S)-PMAL$_m$-b-PSt$_n$ were synthesized by RAFT polymerization and characterized by $^1$H NMR and GPC (see Supplementary Methods and Supplementary Tables 1, 2 for detailed discussion). The chiral structure of amino acid was preserved during the process of polymerization (Supplementary Fig. 4a). The monodisperse $Fe_3O_4$ MNPs with an average diameter of 6 nm were prepared to ensure its superparamagnetic at room temperature for easy recycle (Supplementary Fig. 13)[27]. The block copolymers were designed to have relative short hydrophobic PSt blocks for the formation of micellar structure in water by encapsulating MNPs in the hydrophobic core. The shell of the micelles were further cross-linked by using 2,2′-(ethylenedioxy) bis(ethylamine) as cross-linker and N-ethyl-N'-(3-dimethylaminopropyl)carbodiimide methiodide (EDC) as activator. The obtained nano-splitters were marked as S-Fe-m-n, where S indicates the S configuration of the chiral center in the side chains of PMAL, m represents the number-average degree of polymerization (DP) of PMAL block and n represents the number-average DP of PSt block. Both m and n were varied to change self-assembled morphologies and study their contribution to the resolution effects. The chiral side chain was also changed to R-lysine (the product was named as R-Fe-25–127) for confirming the chiral recognition mechanism. The structure with multiple $Fe_3O_4$ nanoparticles in one micelle was designed, for generating strong magnetic interaction[28]. The number of encapsulated MNPs was varied by changing the weight ratio of copolymer and the MNPs[29]. Superconducting quantum interference device (SQUID) was used to characterize the magnetism of the co-assemblies at 298 K with bare $Fe_3O_4$ as the positive control. The magnetic susceptibility (M)-intensity of magnetic field (H) curves showed that the co-assemblies had a magnetization saturation (Ms) value of 29.1 emu g$^{-1}$, which was lower than the value of 51.3 emu g$^{-1}$ for $Fe_3O_4$ MNPs (Fig. 2a) owing to the introduction of non-magnetic copolymers[30]. No hysteresis was detected in both co-assemblies and $Fe_3O_4$ MNPs. This means the co-assemblies retained superparamagnetic feature of $Fe_3O_4$ MNPs at room temperature. In other words, they would become magnetic only under a magnetic field and thus undesired agglomeration could be avoided in recycling process. The assemblies were well-dispersed in aqueous solution. When a magnetic field with an intensity of 0.2 T was applied, the assemblies separated from the solution in <20 s. The detailed synthesis and characterization of the nano-splitters are shown in Supplementary Methods and Supplementary Figs. 2 and 3.

**Selective crystallization**. We demonstrated our concept by using racemic amino acids (rac-a.a.) as model substrates because of their wide applications. The system with racemic asparagine monohydrate (rac-Asn•$H_2O$) as target compound, worm-like micelles (S-Fe-25–125, Fig. 2b) as nano-splitters, and R-Asn•$H_2O$ as seeds, was taken as the first example. The concentration of rac-Asn•$H_2O$ was fixed to 111 mg mL$^{-1}$, which was a little above the critical boundary of metastable zone at room temperature[12,31] for promoting the nucleation and crystal growth of S-Asn•$H_2O$ after the crystallization of R-Asn•$H_2O$. This kinetic control process regulated by nano-splitters was reflected in the first 24 h of the crystallization process: the colorless R-crystals came out first

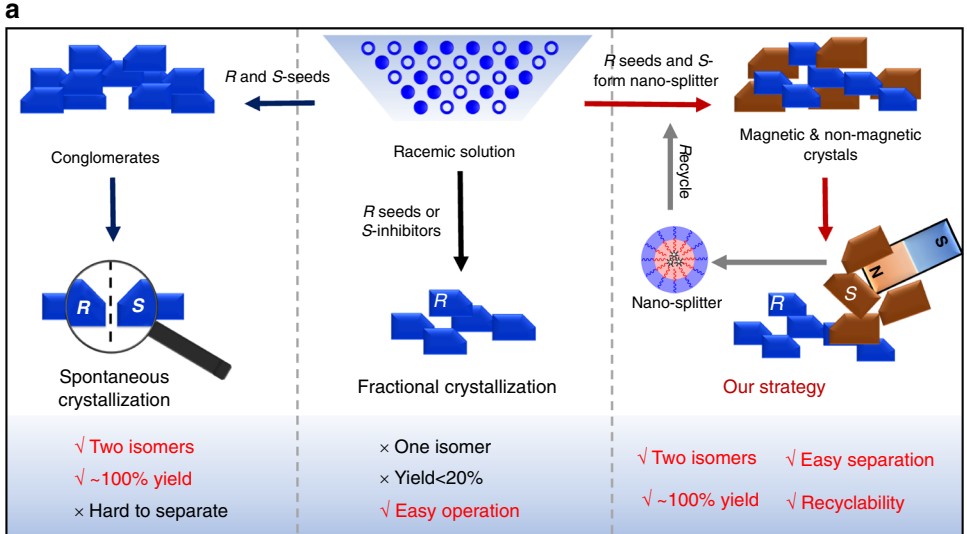

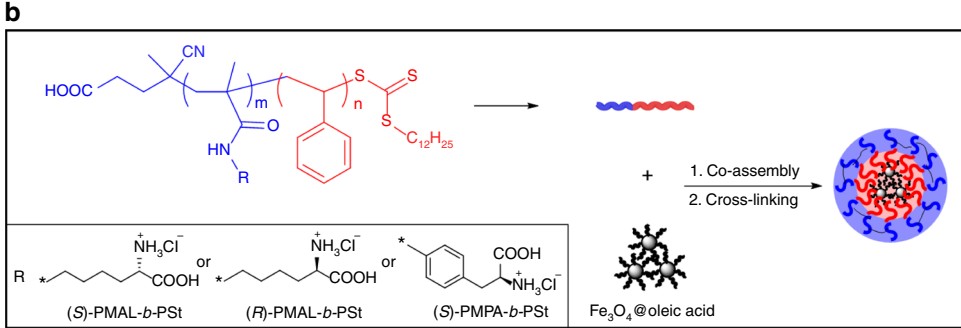

**Fig. 1** Schematic of the magnetic separation system. **a** The concept of our strategy: in the spontaneous crystallization, two isomers with almost 100% yield can be obtained, but the enantiomeric crystals are hard to distinguish. In the fractional crystallization, only one isomer with less than 20% yield can be obtained in one crystallization process. In our strategy, the enantiomeric crystals can be easily separated by a magnetic field with a quantitative yield. **b** The synthetic route of the nano-splitters

and then the dark brown $S$-crystals began to appear. The CD signals of supernatant was consistent with the results of the kinetic experiments (Supplementary Fig. 5). After the solution was stored for 72 h at room temperature without evaporation of solvent, a mixture of dark brown and colorless crystals formed. The dark brown crystals were significantly larger than the colorless ones. They were magnetic and could be easily screened out from the mixture by a permanent magnet (Fig. 2c, and Supplementary Movie 1). As expected, these dark brown crystals consisted of $S$-a.a. with 95.2 ee% according to the chiral HPLC results, while the colorless crystals were $R$-a.a. with 99.1 ee% (Supplementary Fig. 18), the total yield was 39.8%. Moreover, the solvent was slowly evaporated after the brown crystals appeared to generate continual oversaturated state. As expected, a self-sorting process kept going and the enantiomeric crystals grew up without obvious spontaneous nucleation (Fig. 2e). Although the theoretical yield could be 100% by using this method, in practice, completely dry of the solution should be avoided to make sure the crystals didn't adhere to each other. Thus, the total yield was controlled to 95.1% without compromising the ee% values (99.2 ee% for $R$-crystals and 95.0 ee% for $S$-crystals, average of three experiments, Fig. 2e and Supplementary Fig. 19). When $R$-Fe-25–127 and $S$-Asn·H$_2$O seeds were used, similar results were obtained but in which the magnetic crystals were the $R$-a.a., and the non-magnetic crystals were the $S$-a.a. (Fig. 2d and Supplementary Movie 2). It confirmed that the stereoselectivity came from the lysine chiral side groups. In a blank control experiment without nano-splitters, the spontaneous nucleation of $S$-

enantiomer took more time due to the lack of nuclei for the $S$-crystals (Supplementary Fig. 6). When the nanoparticles were added, they would trap the $S$-clusters and lead to a $S$-enatiomer riched area, which in turn accelerate the crystallization of $S$-enatiomer.

## Discussion

The separation efficiency can be tuned by the structure parameters of the $S$-Fe-m-n. In our case, the reason for the impurity of one enantiomer mainly comes from the mixed crystals of the other enantiomer. Therefore, efficient chiral interaction between $S$-nuclei and $S$-PMAL, strong enough magnetization of the nano-splitters, should affect the purity of obtained crystals. Changes in the chain length of $S$-PMAL, morphologies of the co-assemblies, and feeding amount of nano-splitters (number of nucleation centers) were expected to vary the chiral interaction between $S$-nuclei and $S$-PMAL. With increasing the DP of PMAL segments from 12 to 59 at a fixed length of PSt, the best ee% values of obtained enantiomeric crystals were observed in a DP of 25 (Fig. 3a). Short PMAL segments could not provide efficient chiral interaction while longer length of chiral segments led to a smaller number of polymer chains and thus lower effective local concentration at a fixed weight concentration of PMAL segments. On the other hand, we varied the volume fraction of PSt ($\varphi_{PSt}$) to tune the morphology of nano-splitters from spherical/wormlike micelles to vesicles (Fig. 3b, Supplementary Table 3). It was found that the aggregates that provided more contact area with $S$-nuclei

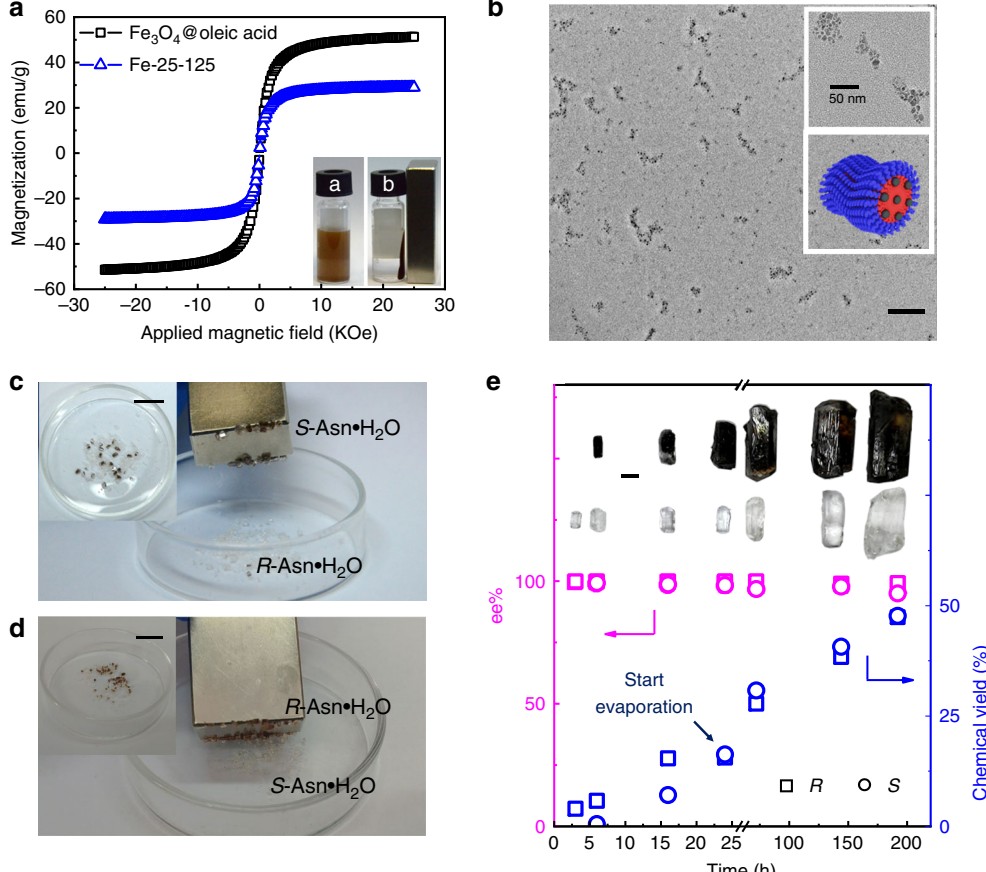

**Fig. 2** Properties of the nano-splitters and their application in selective crystallization. **a** The magnetic hysteresis loops of Fe$_3$O$_4$ MNPs and S-Fe-25–125 (inserted pictures: the aqueous dispersion of magnetic nano-splitters (bottle a) and the separability of the co-assemblies by placing an external magnetic field (bottle b). The time from state a to state b is within 20 s. **b** The TEM images of S-Fe-25–125 (inserted pictures: magnified images (top) and 3D model of the co-assemblies (bottom)). **c** The images of R and S-Asn•H$_2$O crystals by using S-Fe-25–125 as additive. **d** the images of S and R-Asn•H$_2$O crystals by using R-Fe-25–127 as the additive. **e** The variations of chemical yield and ee% of R/S-Asn•H$_2$O crystals over time by using 0.5 wt% of S-Fe-25–125 as the additives for strong magnetic responsiveness of S-crystals, the solvent was slowly evaporated after 24 h (inserted picture: magnified images of the two kinds of crystals growing over time). Scale bars: 100 nm (**a**); 2 cm (**c**); 2 cm (**d**); 1 mm (**e**)

showed better chiral recognition, i.e., wormlike micelles (i.e., S-Fe-25–125) > spherical micelle (i.e., S-Fe-40–122 and S-Fe-59–129) > vesicle (i.e., S-Fe-12–122). Wormlike ones were more flexible such that they could adjust their morphologies to provide larger contact area for more efficient chiral interaction. In addition, we found that increasing the feeding of S-Fe-m-n increased the ee% of R-crystals but decreased that of S-ones (Fig. 3c). Note that here S-Fe-25–174 was used as additive to reveal the changes in ee% for both R and S-crystals, as the ee% for S-crystals was already 99.1% when 0.25 wt% of S-Fe-25–125 was used and no obvious improvement can be achieved when increasing the feeding amount. In a solution with concentrated S-Fe-25–174, small magnetic S-crystals formed due to the relative higher concentration of nucleation centers. As a result, the S-crystals without wrapping any nano-splitters would be much less and their smaller sizes enabled much easier attraction by a magnetic field, making the R-crystals purer. However, the smaller the S-crystal size was, the easier it could adhere to small crystals of the opposite configuration leading to a drop in ee% for S-crystals.

The saturation magnetization of the crystal was varied by the loading amount of Fe$_3$O$_4$ MNPs in the micelle (the average number of Fe$_3$O$_4$ NMPs in spherical micelles was calculated, see detail in Supplementary Equations 1–3 and Supplementary Table 3). With decreasing the loading of MNPs, the micelles became smaller (Fig. 3d) and weak in saturation magnetization

(Supplementary Fig. 14). As a result of the decrease in magnetism, the observed ee% of non-magnetic R-crystals dramatically decreased while that of magnetic S-crystals rose slightly (Fig. 3e). It was attributed to the difficulty in magnetic-induced screen where some S-crystals with weak magnetism could not be collected by a magnet (Fig. 3f–i).

Our strategy was further explored to other conglomerates systems. S-Fe-25–125 was applied to separate Threonine (Thr) and allo-Threonine (aThr), satisfied results were obtained (Fig. 4a and b). When the lysine units in the copolymers were replaced by S-phenylalanine to generate (S)-Fe(PMPA)-45–115 by a similar synthetic protocol (see detail in Supplementary Methods), this tailored nano-splitters could be used to separate the racemic p-hydroxyphenylglycine ptoluenesulfonate (rac-pHpgpTs) (Fig. 4c). The resultant pHpgpTs crystals were small and needle-like. These crystals were pure but slight difficult for screen under magnetic field because of the entanglement of the thin, long crystal fibers. As expected, changing the chiral configuration of side chains led to opposite crystallization order (Fig. 4d).

Taking advantage of magnetic responsiveness, our nano-splitters were easily recycled. We evaluated the recyclability of the magnetic nano-splitters by both their recollection yield and chiral resolution performance with S-Fe-25–125 as an example. During the formation of the enantiomeric crystal, the nano-splitters were entrapped in the S-crystals[26]. They were recollected

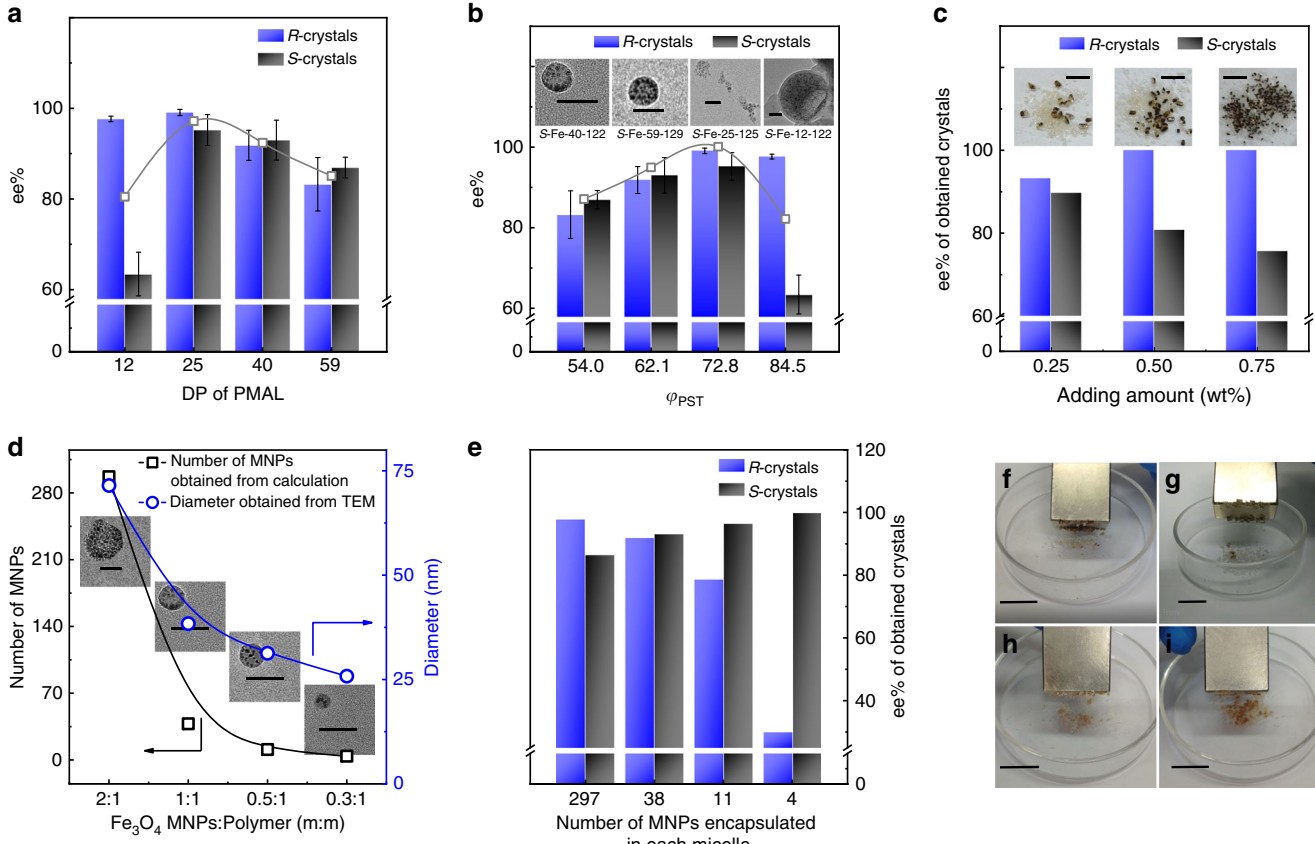

**Fig. 3** Factors that influence the resolution performance. **a** The influence of S-PMAL's DP on the ee% values of the R and S-crystals. The gray curve represents for the average ee% of R and S-crystals. **b** The influence of morphologies on the ee% values of the R and S-crystals (inserted pictures: the typical images of the co-assemblies). The gray curve represents for the average ee% of R and S-crystals. **c** The influence of feeding amount on the ee% values of R- and S-crystals (inserted pictures: the typical images of the crystals), S-Fe-25–174 was used as additive. **d** The particle size and the number of MNPs in each micelle of S-Fe-40–122, x axis represents the weight ratio of MNPs and polymer. **e** The influence of MNPs' number in each micelle of S-Fe-40–122 on the ee% values of the R and S-crystals. **f–i** The images of obtained crystals by using nano-splitters with different sizes (the size decreased from **f–i**), 0.25 wt% of S-Fe-40–122 was used as additive. In all the experiments the concentration of Asn•$H_2O$ was 111 mg mL$^{-1}$, the temperature was 25 °C, the crystallization time was 72 h without evaporation of water. Error bars are standard error of measurement. Scale bars: 50 nm (**b**); 10 mm (**c**); 50 nm (**d**); 2 cm (**f–i**)

by dissolving the magnetic crystals and then being attracted by a magnet (see detail in Methods section and Supplementary Fig. 33). After simply water-washing, the collected nano-splitters could be directly used again. Under the identical crystallization condition, similar ee% and chemical yield were obtained, even after five-time recycling (Fig. 4e, f), indicating excellent recyclability and stability. Moreover, the recovery ratio of each cycle kept around 98%. Considering the weight loss in washing, the recovery ratio was nearly quantitative.

In summary, we have reported a strategy for resolution of conglomerates involving a magnetic separation process with the assistance of magnetic nano-splitters. This kind of nanomaterial is built on the co-assembly of MNPs and amphiphilic polymeric tailor-made additives. The nano-splitters lead to distinguish apparent properties of enantiomeric crystals, including magnetism, size, and color. Thus, easy and quantitative isolation of the R- and S-crystals in a single crystallization process can be realized. Compared to traditional methods, this strategy doesn't require special equipment or any chemical transformation process. Moreover, nano-splitters can be easily recovered and reused without compromising their separating capability. Although this strategy only works in the case of conglomerates at this stage, we still envision that our strategy opens a window to develop additives for different scale chiral resolution.

## Methods

**Synthesis of $Fe_3O_4$ nanoparticles.** The $Fe_3O_4$@oleic acid was prepared according to literatures:[32,33] benzyl ether (150 mL), Fe(acac)$_3$ (5.3 g, 15.0 mmol), 1,2-hexadecanediol (19.4 g, 75.1 mmol), oleylamine (12.0 g, 45.0 mmol) and oleic acid (12.7 g, 45.0 mmol) were added into a flask and well stirred under nitrogen. The mixture was heated to 200 °C for 2 h, under the protection of nitrogen, and then heated to 300 °C for 1 h. After cooled to room temperature, ethanol was added into the dark-brown mixture under air, and dark-brown particles were precipitated. After centrifugation, the product was dispensed in hexane in the presence of oleic acid and oleylamine and reprecipitated with ethanol to give 6 nm $Fe_3O_4$ nanoparticles.

**Co-assembly of $Fe_3O_4$@oleic acid and diblock copolymers.** Fe-m-n were prepared according to literature:[29] To encapsulate $Fe_3O_4$ nanoparticles within PSt-b-PMAL/PSt-b-PMPA, freshly prepared 4 mL PSt-b-PMAL/PSt-b-PMPA stock solution (10 mg mL$^{-1}$ in DMSO) was diluted with a mixture of 36 mL of DMSO. To this solution 40 mL of 6 nm $Fe_3O_4$@oleic acid nanoparticle stock solution (1.0 mg mL$^{-1}$ THF) was added slowly with vigorous stirring, such that [$Fe_3O_4$]$_{initial}$ = 0.5 mg mL$^{-1}$ in 50:50 DMSO/THF, [copolymer]$_{initial}$ = 0.50 mg mL$^{-1}$ in 50:50 DMSO/THF. Then, 320 mL of $H_2O$ was gradually added dropwise to the solution at a rate of 0.5 mL min$^{-1}$ with vigorous stirring.

The PMAL/PMPA blocks of the resulting micelles were cross-linked by adding 2,2′-(ethylenedioxy)bis(ethylamine) as a difunctional linker and EDC as an activator. For 30% crosslinking of the PMAL segments, 1.44 mL of freshly prepared EDC solution (1.0 wt% in DMF) was added into the assemblies' solution. The resulting suspension was left for activation for 30 min. Followed by adding 0.7 mL 2,2′-(ethylenedioxy)bis(ethylamine) solution (1.0 wt% in DMF) and stirred overnight.

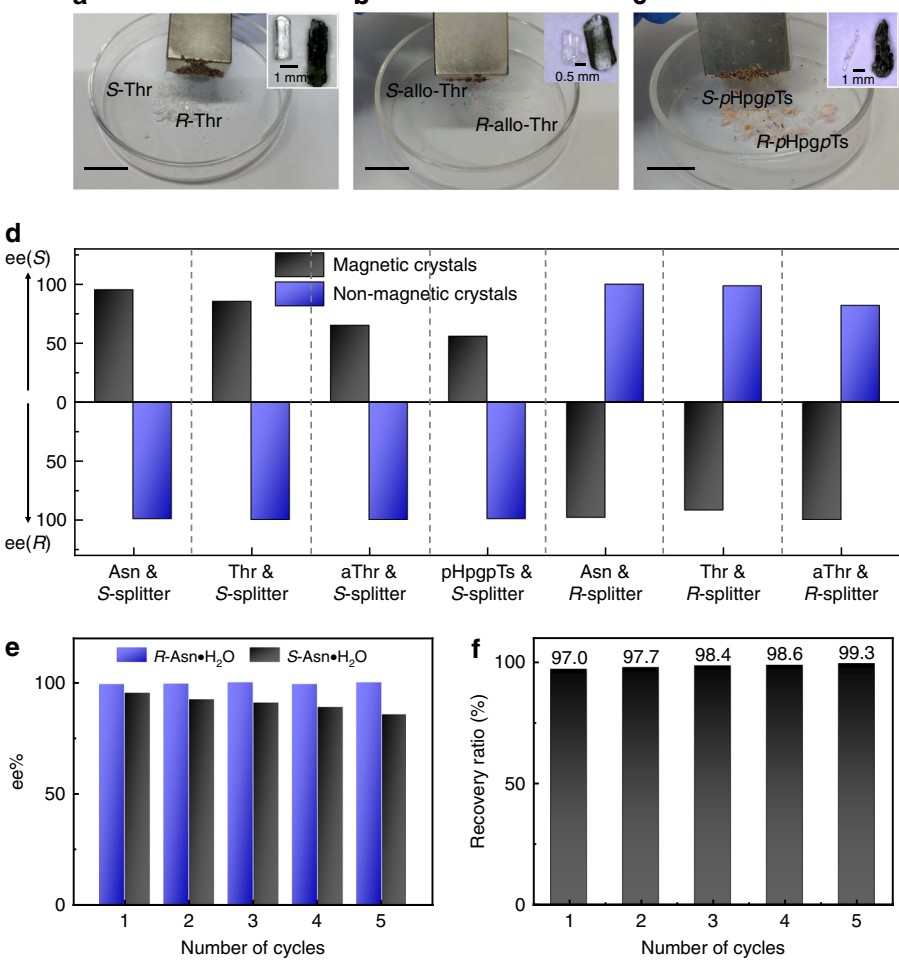

**Fig. 4** Extension of this strategy and the recyclability. **a** The typical photo of $S$ and $R$-Thr crystals in the present of $S$-Fe-25–125. **b** The typical photo of $S$ and $R$-aThr in the present of $S$-Fe-25–125. **c** The typical photo of $S$ and $R$-$p$Hpg$p$Ts in the present of $S$-Fe(PMPA)-45–115. **d** The ee% values for both $R$ and $S$-crystals in different systems. **e** The crystallization results by using recycled $S$-Fe-25–125. **f** The recovery ratio of $S$-Fe-25–125 for each cycle. Scale bars: 2 cm (**a**–**c**)

The resulting suspension was subjected to the dialysis against distilled water for over 72 h (Genia Biotech® Regenerated Cellulose Membrane, MWCO = 3500 Da) to remove any residual solvent and unreacted small molecules. The micelle solution was subjected to centrifugation for 30 min. The upper 90% of the solution was discarded and the same volume of water was added to the solution. This procedure was repeated three times. The residual liquid was freeze-dried to obtain brown powders. Purified nano-splitters were readily redispersed in distilled water by ultrasonic dispersion. Two microliters of the diluted solution was deposited on the TEM grid, dried in air before taking TEM images.

**Selective crystallization of *rac*-Asn•H₂O**. In a typical process by using *rac*-Asn•H₂O as substrates and *S*-Fe-25–125 as additives, the supersaturated solution (1.0 g of *rac*-Asn•H₂O in 9 mL H₂O) was heated at 60 °C until complete dissolution occurred, filtered, and then 5 g supersaturated solution was transferred to a hot penicillin bottle, a certain amount of *S*-Fe-25–125 was added in and dispersed by ultrasound in 50 °C water bath. The whole solution was gradually cooled down to 25 °C. After being left stand at 25 °C for 30 min, seeds of *R*-Asn•H₂O were added in. After 72 h, the supernatant fluid was collected and the formed crystals were washed with ethanol/water (v/v, 50/50) × 3, and then dried under vacuum. For the 95% chemical yield, the solvent was evaporated slowly at 28 °C with gentle stirring after the brown magnetic crystals were formed.

**Selective crystallization of *rac*-Thr**. In a typical process by using *rac*-Thr as substrates and *S*-Fe-25–125 as additives, the supersaturated solution (2.7 g of *rac*-Thr in 9 mL H₂O) was heated at 65 °C until complete dissolution occurred, filtered, and then 2.2 g supersaturated solution was transferred to a hot penicillin bottle, a certain amount of *S*-Fe-25–125 was added in and dispersed by ultrasound in 50 °C water. The whole solution was gradually cooled down to 25 °C. After being left stand at 25 °C for 30 min, seeds of *R*-Thr were added in. After 120 h, the

supernatant fluid was collected and the formed crystals were washed with ethanol/water (v/v, 50/50) × 3, and then dried under vacuum.

**Selective crystallization of *rac*-aThr**. In a typical process by using *rac*-aThr as substrates and *S*-Fe-25–125 as additives, the supersaturated solution (1 g of *rac*-aThr in 7.5 mL H₂O) was heated at 65 °C until complete dissolution occurred, filtered, and then 4.2 g supersaturated solution was transferred to a hot penicillin bottle, a certain amount of *S*-Fe-25–125 was added in and dispersed by ultrasound in 50 °C water. The whole solution was gradually cooled down to 25 °C. After being left stand at 25 °C for 30 min, seeds of *R*-aThr were added in. Then the whole solution was left stand at 10 °C to crystallize. After 72 h, the supernatant fluid was collected and the formed crystals were washed with ethanol/water (v/v, 50/50) × 3, and then dried under vacuum.

**Selective crystallization of *rac*-$p$Hpg$p$Ts**. In a typical process by using *rac*-$p$Hpg$p$Ts as substrates and *S*-Fe(PMPA)-25–125 as additives, the supersaturated solution (2.5 g of *rac*-$p$Hpg$p$Ts in 10 mL 0.5 M $p$-Ts solution) was heated at 60 °C until complete dissolution occurred, filtered, and then 2.5 g supersaturated solution was transferred to a hot penicillin bottle, a certain amount of Fe(PMPA)-25–125 was added in and dispersed by ultrasound in hot water. The whole solution was gradually cooled down to 25 °C (2 °C per 10 min). After being left stand at 25 °C for 30 min, seeds of *R*-$p$Hpg$p$Ts were added in. After a period of time, the temperature was raised to 30 °C to allow ripening. Finally, the supernatant fluid was collected and the formed crystals were washed with cold acetone, and then dried under vacuum.

**Recycling of the nano-splitters**. The nano-splitters in supernatant fluid were recollected by using a magnet. The nano-splitters in *S*-crystals were recollected by

dissolving the crystals and attracted by a magnet. The recollected nano-splitters were washed with distilled water, and then freeze-dried under vacuum.

## Data availability

The data that support the findings of this study are available from the corresponding author upon request.

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

## Acknowledgements

We gratefully acknowledge the financial supports from the National Natural Science Foundation of China (Nos. 51833001 and 21674002). J.X.C. acknowledges the support from BMBF under the project of the Leibniz Research Cluster with an award number 031A360D. We thank Prof. Jinying Yuan and Dr. Qiquan Ye for their help with the TEM measurements. We thank Dr. Jin Xiong for his help with magnetization curve measurements.

## Author contributions

X.C.Y., J.Z., and X.H.W. contributed to the conception; X.C.Y., J.X.C., and X.H.W. contributed to design of experiments, drafting and critical revision of the manuscript; X.C.Y., and B.W.L. contributed to synthesis, analysis and data collection; N.L. contributed to the design of copolymers; R.W. and J.Y.T. contributed to the measurement of TEM; Z.J.Y. contributed to the polymerization.

## Additional information

**Competing interests:** The authors declare no competing interests.

