## [Peer Review File · Nature Communications]

Reviewers' comments:

Reviewer #1 (Remarks to the Author):

This paper is a very nice contribution to the field of chiral resolution. Indeed for conglomerates separating R- from the S-enantiomer, often requires tedious and repeated processes, which are industrially not always easy to set up.

The authors present a novel conglomerate separation methodology by magnetizing the crystals of only one of the two enantiomers of a given system.

I feel this paper is indeed of interest to a more general public and will open up future opportunities for academic as well as industrial challenges related to chiral resolution. In particular, I feel that pharmaceutical industry might find this approach interesting. The technique presented possibly opens to other systems (not only conglomerates) and I believe it will be rapidly used by different researchers.

The claims are really carefully explained and the experiments illustrated clearly.

The data treatment is ok and the authors clearly do not oversell.

However, at this stage I would not recommend it for acceptance but rather recommend a re-submission after the following points have clearly been dealt with.

1. There is a serious need for revising the English. I feel that I needed to reread a lot of the sentences prior to understanding, and this implies that a general public is likely going to have to do the same. People are not going to do this until the end of the paper, and I feel that the authors will gain much less impact than what this work can achieve. I would therefore strongly recommend the authors to revisit the English thoroughly and in particular take care of cutting some of the longer sentences so that only one main idea is presented in each sentence. Maybe asking the help of a native English speaker would definitely benefit this paper, especially because I feel that it is interesting.

2. The authors should also in the introduction clearly highlight that this paper orients towards 'Separation' and make a distinction towards 'transformation'. Processes such as Viedma Ripening and Crystallization Induced Asymmetric Transformation, transform a racemic mixture into an enantiopure compound and as such lead to enantiopure crystals in a single process (contrary to their claim). I agree with the authors that for separation processes of conglomerates the processes are often tedious, but not for the transformation processes. This distinction should clearly be made. The value of the process here is that it does not require the target compound to be racemizable.

3. The authors should also clearly state the limitations of their work which according to me is the application to conglomerates. These represent 10% of compounds, and therefore the authors should clearly state this. Maybe there is a way of expanding this technique to other systems (and I strongly believe so) but this should be noted.

4. What I believe to be strong-point is the fact that the SPLITTERS come off with the crystals that do not have the optimal ee. The ones remaining in solution do have a very impressive ee, so this shows the value of the technique. The authors furthermore show how to remove and re-use the SPLITTERS. What I would nevertheless like to see added to the paper, or commented upon concerns the interaction between the SPLITTERS and the crystals being removed. Does the splitter absorb only on the surface or is the SPLITTER incorporated (eg through vesicles or other) in the crystalline material. In other words, the crystals are brown. Is this a mere surface effect or not? In the former case, a simple slight first washing should lead to full removal of this brown color.

5. I wonder if the authors are not able to make this a more shorter communication even, by placing one or two more paragraphs into the supporting. But this I think might become clear upon revising.

So to summarize, I do feel that the data is extremely interesting and definitely worth being published but the authors should first review this manuscript and resubmit it.

Tom Leysens

Reviewer #2 (Remarks to the Author):

In this, paper Xinhua Wan et.al present their results on a new strategy for quantitatively chiral separating conglomerates crystals using magnetic nanoparticles. The motivation of this research is quite interesting and the subject itself is very relevant to chiral chemistry. Overall this work is very original and in my opinion, the results presented in this study are a significant chiral resolution by crystallization. This is a well-written article with comprehensive experimental results, (this is a good example for rigorous scientific study). I can only come up with a few arguments that I think the authors should correct:

1) Indeed, as the authors claim chiral separation by classic selective crystallization is very of importance, however, the authors must emphasize that crystallization of chiral system as conglomerates are very rare only about 7 % of all chiral crystals crystallized as chiral conglomerates.

2) One key and important question relates to the chirality of the magnetic nanoparticles, is the chirality preserved during the process of polymerization? I think the authors need to present the results of optical measurements (CD - circular dichroism spectroscopy) of polymeric magnetic nanoparticles.

3) Another important point is the effect of the "magnetic splitters" on the crystallization process,, it is not clear to me how the chiral magnetic splitters influence the crystallization,? I think that time-resolved spectroscopy (e.g CD or polarimetry) of the crystallization with and without the magnetic splitters can greatly contribute to our understanding the mechanism of action of the "magnetic splitters: on the chiral crystallization.

4) One point the authors need to explain is how the crystals+ magnetic splitters can be separated for at the end of the chiral crystallization in order to obtain the pure enantiomer?

In conclusion, I recommend the publication of this paper after the authors' response to the comments raised in this report.

Reviewer #3 (Remarks to the Author):

Ye et al. present in this paper an original method to separate conglomerates, driven by the separation of one enantiomer by magnetic separation after enantioselective crystallization with magnetic seeds. The problem of separating enantiomeric crystals in a conglomerate is solved by using hybrid structures based on magnetite nanoparticles and amphiphilic block copolymers, which serve as crystallization seeds. Overall, I find the work highly interesting, but I have a few concerns that should be addressed by the authors before the manuscript can be accepted for publication.

1. Minor stylistic aspects and typos:

a) p. 1, "discovery of the chiral structure in 1848"  rephrase "the chiral structure", since Pasteur did not discover "the chiral structure" as such

b) p. 5, line 6 from bottom: "was also changed" instead of "wa also changed"

c) p. 12: "involving a MAGNETIC BENEFICIATION LIKE process"  please consider rephrasing this phrase

2. The term "SPLITers" appears to me as a very forced and with very little meaning: what does "stereoselective-inhibition promoted local isomer trappers" exactly mean? I would personally avoid such artificial terms and find a more straightforward term, even if less fancy.

3. Taking into account the analogous character that the hybrid seeds used in this paper can have with other colloidal nanoscopic structures, I miss the reference to previous works dealing with the application in enantioselective crystallization of amino acids of chiral polymer nanoparticles prepared by colloidal methods (e.g., emulsion methods).

4. Figure 2(b): the authors justify the lower magnetization value in emu/g of the hybrid structure with respect to the magnetite nanoparticles by the "shielding effect of the polymeric shell". What is

the content in magnetite of the hybrid structures? The difference in the magnetization seems to me to be related to the content of magnetite itself and not to any "shielding effect". If the magnetite content is lower, the magnetization value will also be lower.

Replies for Reviewer 1:

- (1) There is a serious need for revising the English. I feel that I needed to reread a lot of the sentences prior to understanding, and this implies that a general public is likely going to have to do the same. People are not going to do this until the end of the paper, and I feel that the authors will gain much less impact than what this work can achieve. I would therefore strongly recommend the authors to revisit the English thoroughly and in particular take care of cutting some of the longer sentences so that only one main idea is presented in each sentence. Maybe asking the help of a native English speaker would definitely benefit this paper, especially because I feel that it is interesting.

Thanks for your careful review and helpful advice. We have revised the paper thoroughly and carefully. Some mistakes in statements have been corrected and some long sentences have been cut into short ones. But it is difficult for us to invite a native English speaker to rewrite our paper in a limited time.

Major revises are highlighted in blue and listed here: lines 1-3, page. 2; lines 7-9, page. 2; lines 6-8, page. 3; lines 24-26, page. 7; lines 14-16, page. 8; lines 2-3, page. 10; lines 9-15, page. 12.

- (2) The authors should also in the introduction clearly highlight that this paper orients towards 'Separation' and make a distinction towards 'transformation'. Processes such as Viedma Ripening and Crystallization Induced Asymmetric Transformation, transform a racemic mixture into an enantiopure compound and as such lead to enantiopure crystals in a single process (contrary to their claim). I agree with the authors that for separation processes of conglomerates the processes are often tedious, but not for the transformation processes. This distinction should clearly be made. The value of the process here is that it does not require the target compound to be racemizable.

We appreciate this insightful comment. We agree that the transformation processes of racemizable compounds is simple in operation. We have revised the statement to distinguish the difference from "separation" to "transformation".

The revised statement can be seen in line 10-19, page. 2.

- (3) The authors should also clearly state the limitations of their work which according to me is the application to conglomerates. These represent 10% of compounds, and therefore the authors should clearly state this. Maybe there is a way of expanding this technique to other systems (and I strongly believe so) but than this should be noted.

Thank you for this insightful comment! we have clearly discussed the limitations of our work.

The revised statement can be seen in lines 17-18, page. 12.

- (4) What I believe to be strong-point is the fact that the SPLITERS come of with the crystals that do not have the optimal ee. The ones remaining in solution do have a very impressive ee, so this shows the value of the technique. The authors furthermore show how to remove and re-use the SPLITERS. What I would nevertheless like to see added to the paper, or commented upon concerns the interaction between the SPLITERS and the crystals being removed. Does the splitter absorb only on the surface or is the SPLITER incorporated (eg through vesicles or other) in the crystalline material. In other words, the crystals are brown. Is this a mere surface effect or not? In the former case, a simple slight first washing should lead to full removal of this brown color.

We believe that the SPLITERS not only incorporated in the crystals but also absorbed onto the surface. It has been discussed briefly in the revised manuscript (lines 1-3, page 12). We have discussed this incorporating behavior in detail in our previous paper (*Angew. Chem. Int. Ed.* **2018**, *130*, 8252-8256). In the previous work, we synthesized a kind of polymer inhibitors to delay the crystallization process of one of enantiomers, leading to subsequent formation of two amino acid (a.a.) enantiomeric crystals. By using a dye to label the polymer inhibitor, we found that in the enantiomeric crystal formed later, the inhibitors concentrated on both the core and the surface of the dyed crystal. We proposed that the formation of the dyed crystal involved an enriching process of a.a. clusters on the polymeric assemblies and an Ostwald ripening process. The a.a. clusters, which were smaller than the critical nucleus size, were entrapped by the polymer inhibitor. An Ostwald ripening process should occur in the cluster enriched polymeric assemblies, which led to a high local a.a. concentration to trigger nucleation. After the dyed core formed, small a.a. molecules, rather than inhibitors, selectively deposit on the growing crystal planes, leading to the dye-free middle region. The inhibitors finally attached to the surface of grown crystals when no excess a.a. molecules were available in the solution. Similar process should occur in the separation described in the current manuscript.

In our additional experiment, we found that it is difficult to remove the SPLITERS absorbed on the surface by simply washing with water, due to the strong interaction (hydrogen bond) between SPLITERS and crystal (Supplementary Fig. 32). Only in the case that the magnetic crystals were dissolved in water, the SPLITERS were released from the crystals.

- (5) I wonder if the authors are not able to make this a shorter communication even, by placing one or two more paragraphs into the supporting. But this I think might become clear upon revising.

We have tried our best to make the paper more concise.

Replies for Reviewer 2:

- (1) Indeed, as the authors claim chiral separation by classic selective crystallization is very of importance, however, the authors must emphasize that crystallization of chiral system as conglomerates are very rare only about 7 % of all chiral crystals crystallized as chiral conglomerates.

Thanks for insightful comment! We have emphasized in the revised manuscript that our method is only work for conglomerates.

The revised statement can be seen in lines 17-18, page. 12.

- (2) One key and important question relates to the chirality of the magnetic nanoparticles, is the chirality preserved during the process of polymerization? I think the authors need to present the results of optical measurements (CD - circular dichroism spectroscopy) of polymeric magnetic nanoparticles.

We have conducted additional experiment based on your suggestions. The CD spectra of *S*- and *R*-PMAL have been shown in Figure R1 (A). The chiral structure of amino acid was preserved during the process of polymerization. Similarly, both *S*- and *R*-nano-splitters show cotton effect, indicating that the chirality has been preserved in self-assembly.

The related figures (Supplementary Fig. 3, in the revised support information) and discussion have been added into revised manuscript (line 3, page. 5).

Figure R1. CD spectra and UV-vis spectra of PMAL and nano-splitters. A. CD spectrum (up) and UV-vis spectrum (down) of the solution of *R/S*-PMAL (DP=25), C (*R/S*-PMAL) = 0.01 mg•mL⁻¹; B. spectrum (up) and UV-vis spectrum (down) of *R/S*-nano-splitters (KBr plate).

- (3) Another important point is the effect of the “magnetic splitters“ on the crystallization process,, it is not clear to me how the chiral magnetic splitters influence the crystallization,? I think that time-resolved spectroscopy (e.g CD or polarimetry) of the crystallization with and without the magnetic splitters can greatly contribute to our understanding the mechanism of action of the “magnetic splitters: on the chiral crystallization.

Thank you for this insightful suggestion! We have monitored the crystallization process in the presence/absence of splitters by CD spectroscopy. The supernatant formed at different time was collected and filtrated out insolubles for CD measurement. The results can be seen in Figure R2. In the presence of *S*-nano-splitters, the positive cotton effect decreases over 24 hours and even became negative at 24 h, which is consistent with the results of the kinetics experiments (Figure 2d). In the absence of nano-splitters, the decrease in CD signal is slower.

In the absence of splitter, the spontaneous nucleation of *S*-enantiomer should take more time due to the lack of nuclei for the *S*-crystals. When the nanoparticles were added, they would trap the *S*-clusters and lead to a *S*-enantiomer riched area, which in turn accelerate the crystallization of *S*-enatiomer.

The related figures (Supplementary Fig. 4 and Fig. 5, in the revised support information) and discussion have been added into revised manuscript (lines 17-18, page. 7; lines 8-12, page. 8).

Figure R2. CD spectra and UV-vis spectra of the supernatant over time. A. CD spectrum (up) and UV-vis spectrum (down) of the supernatant when *S*-nano-splitter and *R*-seeds were added; B. CD spectrum (up) and UV-vis spectrum (down) of the supernatant when only *R*-seeds were added.

Figure R3. CD signals at 200 nm over time.

- (4) One point the authors need to explain is how the crystals+ magnetic splitters can be separated for at the end of the chiral crystallization in order to obtain the pure enantiomer?

The nanoparticles in supernatant fluid were recollected by using a magnet, and the nanoparticles in magnetic crystals were recollected by dissolving the crystals and attracted by a magnet. The recollected nanoparticles were washed with deionized water, and then freeze-dried under vacuum. We have described this process in the supplementary information (see Supplementary Section 1.3 and Supplementary Fig. 32). We have briefly described it in the revised main text for better reading.

The revised statement can be seen in lines 1-3, page. 12.

Replies for Reviewer 3:

(1) Minor stylistic aspects and typos:

- a) p. 1, "discovery of the chiral structure in 1848"  rephrase "the chiral structure", since Pasteur did not discover "the chiral structure" as such
- b) p. 5, line 6 from bottom: "was also changed" instead of "wa also changed"
- c) p. 12: "involving a MAGNETIC BENEFICIATION LIKE process"  please consider rephrasing this phrase

Thank you very much for your advice. We have corrected the mistakes you have mentioned.

- (a) line 3, page. 1, "discovery of the chiral structure" was changed to "Pasteur's separation of sodium ammonium tartrate".
- (b) line 14, page. 5, "wa also changed" was changed to "was also changed".
- (c) line 10, page. 12, "involving a magnetic beneficiation like process" was changed to "involving a magnetic separation process".

(2) The term "SPLITers" appears to me as a very forced and with very little meaning: what does "stereoselective-inhibition promoted local isomer trappers" exactly mean? I would personally avoid such artificial terms and find a more straightforward term, even if less fancy.

We have replaced the artificial term "SPLITer" with nano-splitter to describe this kind of nanomaterials.

The revised statement can be seen in lines 5-7, page. 1.

(3) Taking into account the analogous character that the hybrid seeds used in this paper can have with other colloidal nanoscopic structures, I miss the reference to previous works dealing with the application in enantioselective crystallization of amino acids of chiral polymer nanoparticles prepared by colloidal methods (e.g., emulsion methods).

Thank you for kindly reminding! We have added a reference about the chiral polymer nanoparticles prepared by colloidal methods used for enantioselective crystallization.

See ref. [13] Preiss, L. C. et. al. Amino-acid-based chiral nanoparticles for enantioselective crystallization, *Adv. Mater.* **27**, 2728-2732 (2015).

(4) Figure 2(b): the authors justify the lower magnetization value in emu/g of the hybrid structure with respect to the magnetite nanoparticles by the "shielding effect of the polymeric shell".

What is the content in magnetite of the hybrid structures? The difference in the magnetization seems to me to be related to the content of magnetite itself and not to any "shielding effect". If the magnetite content is lower, the magnetization value will also be lower.

Thank you for your kindly reminding! We agree that the magnetization is related to the content of magnetite itself.

The content of magnetite can be calculated from TGA test (Supplementary Fig. 6). Taking *S*-Fe-25-125 shown in figure 2b as an example, the mass fraction of Fe₃O₄ is 42.4%. Therefore, the magnetization of *S*-Fe-25-125 is expected to be about half of Fe₃O₄@oleic acid.

The revised statement can be seen in lines 5-6, page. 7.

In addition, all the editorial points have been addressed according to the requirement. Thank you again for your time and consideration. We are looking forward to your positive response.

With all my best wishes

Yours sincerely,

Xinhua Wan

February 24, 2019

REVIEWERS' COMMENTS:

Reviewer #1 (Remarks to the Author):

The authors clearly took into consideration all the comments and furthermore performed supplementary experiments.

This has strongly reinforced the article which I know recommend for publication.

Tom Leysens

Reviewer #2 (Remarks to the Author):

In the revised paper, the authors answered in an appropriate way and responded to all claims raised by the reviewers. Moreover, the authors conducted more experiments following the recommendations of the reviewers . The article now can be published in its current form.

Reviewer #3 (Remarks to the Author):

In my opinion, the authors have addressed correctly the comments and concerns of the reviewers. The manuscript can be now accepted upon discretion of the editor.